

# 1 3D Seismic Traveltime Tomography Validation of a Detailed
# 2 Subsurface Model: The case study of the Zancara River Basin
# 3 (Cuenca, Spain)

David Marti[1], Ignacio Marzan[1], Jana Sachsenhausen[1], Joaquina Alvarez-Marron[1], Mario Ruiz[1],
Montse Torne[1], Manuela Mendes[2] and Ramon Carbonell[1]
[1] Institute of Earth Sciences Jaume Almera, ICTJA, CSIC, Lluis Solé I Sabaris s/n, 08028, Barcelona, Spain
[2] Department of Physics, Instituto Superior Técnico, IST, Av. Rovisco Pais, 1049-001 Lisbon, Portugal
*Correspondence to: David Marti dmarti@ictja.csic.es*
**ABSTRACT**
A high-resolution seismic tomography survey was acquired to obtain a full 3D P-wave seismic velocity image in the
Zancara River Basin (east of Spain). The study area consists of lutites and gypsum from a Neogene sedimentary
sequence. A regular and dense grid of 676 shots and 1200 receivers was used to image a 500x500 m area of the shallow
subsurface. A 240-channel system and a seismic source consisting of an accelerated weight drop, were used in the
acquisition. Half million traveltime picks were inverted to provide the 3D velocity model that allowed to resolve the
structure up to 120 m depth. The project targeted the geometry of the underground structure with emphasis in defining
the lithological contacts but also the presence of cavities and fault/fractures. An extensive drilling campaign provided
uniquely tight constraints on the lithology; these included core samples and wireline-log geophysical measurements.
The analysis of the well-log data enabled the accurate definition of the lithological boundaries and provided an estimate
of the seismic velocity ranges associated to each lithology. The final joint interpreted image reveals a wedge shaped
structure consisting of four different lithological units. This study features the necessary key elements to test the
traveltime tomographic inversion approach in the high-resolution characterization of the shallow subsurface. In this
methodological validation test, traveltime tomography demonstrates to be a powerful tool with a relatively high
capacity for imaging in detail the lithological contrasts of evaporitic sequences located at very shallow depths, when
integrated with additional geological and geophysical data.
**1. INTRODUCTION**
Knowledge of the very shallow structure of the Earth has become a critical demand for the modern society. The shallow
subsurface is the part of the Earth with which humans have the most interaction. Characterizing the subsurface is
important since it hosts critical natural resources; it is used as reservoir for resources and waste, plays a key role in
support of infrastructure planning and holds the imprint of the anthropogenic processes. Thus, understanding its
composition and structure is a regular objective in studies such as: natural resource exploration [Davis et al., 2003;
Place et al, 2015] and environmental assessment studies [Steeples, 2001; Zelt et al, 2006)] It is also critical in civil
engineering practice and monitoring of underground structures [Escuder-Viruete et al., 2003; Malehmir et al., 2007;
Juhlin et al., 2007; Martí et al., 2008; Giese et al., 2009; Alcalde et al., 2013a]. In addition, the implementation of a
competent subsurface exploration scheme is very valuable for assessing and/or providing detailed site characterization
for addressing natural hazards, e.g., seismic hazard [Samyn et al. 2012; Ugalde et al., 2013; Wadas et al., 2017; Bernal
et al., 2018]. Typical geotechnical practice for subsurface exploration has often relied on a combination of drilling, in
situ testing, geophysical surveys, and laboratory analysis of field samples [Andara et al., 2011; Kazemeini et al., 2010;
Alcalde et al., 2014].
Geophysical techniques provide a great variety of approaches to accurately describe the structure, and the distribution of
different physical properties in the subsurface. Depending on the target depth and the required spatial resolution
different methodologies can be applied [e.g. Bryś et al., 2018; Novitsky, et al., 2018; Malehmir et al., 2009 and 2011;
Escuder-Viruete et al 2004; Carbonell et al., 2010; Ogaya et al., 2016; Andres et al. 2016, Alcalde et al., 2013b]. Since
the late 90's sophisticated geophysical techniques have been developed to estimate near-surface velocity models as a
proxy for subsurface stiffness in seismic applications with different targets [Bergman et al., 2004 and 2006; Heincke et



al., 2010]. Seismic traveltime tomography is a robust, efficient and well contrasted tool to constrain the rock's physical properties at very shallow depths [Yordkayhun et al., 2009b; Flecha et al., 2004; 2006; Marti et al., 2002a; ; Baumann-Wilke et al., 2012]. When seismic data is densely acquired it can provide very high spatial resolution images even in 3D at a much affordable cost than conventional 3D seismic reflection surveys. In areas where the geology has not particularly internal structural complexity and with moderate lateral lithological variability, traveltime tomography can provide a reliable image of the subsurface [Marti et al., 2002b; 2006; Yordkayhun et al., 2009a; Letort et al., 2012; Baumann-Wilke et al., 2012]

The study area, in the Loranca Bansin (Cuenca, Spain), has been considered as a possible host for a singular facility for temporary storage of radioactive waste. The emplacement of such a facility requires an extensive multi-scale, multi-disciplinary knowledge of the site's subsurface [Witherspoon et al., 1981; IAEA 2006; Kim et al, 2011], including a detailed 3D image of the structure and the distribution of its physical properties, specially focused on the upper hundred meters which directly interact with the ongoing construction works. The available data suggests that the sedimentary sequence in the study area presents certain tilting to the west, with no significant faulting and therefore no great structural complexity is expected in the shallow subsurface. Following similar case studies on very shallow site characterization, traveltime tomography was considered as the most adequate geophysical method to provide constraints on the seismic velocities for the 3D baseline model [Martí et al., 2002b; Juhlin et al., 2007; Yordkayhun et al., 2007].

A very dense source-receiver grid was designed to assure the necessary lateral resolution and depth coverage of the seismic data, to constrain the geological features of interest beneath the construction site. The specific target was to resolve in detail the internal geometry of a mostly gypsiferous succession with diffuse lithological boundaries [Martinius et al., 2002; Diaz-Molina and Muñoz-Garcia, 2010; Escavy et al., 2012;]. The inversion of almost half a million first arrival traveltime picks provided a 3D distribution of the P-wave velocities. This combined with borehole information allowed to fully characterize three main lithological units and constrain the interpretation of the velocity model. Borehole information was instrumental to define the specific 3D geometry of the different lithologies in the tomographic model and, the topography of the complex boundaries.

## 2. GEOLOGICAL SETTING

The area of Villar de Cañas (Cuenca, Spain) is included in the Loranca Basin, in the southwestern branch of the Iberian Chain (Fig. 1a). The Iberian Chain corresponds to a wide mostly east-southeast trending Alpine intraplate orogen in the eastern Iberian Peninsula. . The structure consists of a thin-skinned, west-verging, mostly imbricate thrust system and associated fault-propagation folds that deform a Mesozoic and Cenozoic sedimentary cover detached above the Paleozoic basement [Muñoz-Martín y De Vicente, 1998; Sopeña y De Vicente, 2004]. The thrust faults merge at depth into a basal detachment located within Middle-Upper Triassic sequences [Piña-Varas el al, 2013].

The crustal structure of the Iberian Chain has gathered academic interest since the early 1990's [see Seille et al, 2015; Guimerà et al., 2016 and references therein], including the acquisition of local and regional geologic and geophysical studies of the Loranca Basin [Biete et al., 2012; Piña-Varas et al., 2013]. The Loranca Basin comprises syntectonic Cenozoic strata [Guimerà et al., 2004]. It has been interpreted as a piggy-back basin that evolved during the Late Oligocene-Early Miocene period and includes mostly fluvial and lacustrine facies sediments, organized into three major alluvial fan sequences and their associated flooding plains (Diaz-Molina and Tortosa, 1996). The alluvial fans were fed from the southeastern and western boundaries of the basin and were comprised of mostly sandstones, gravels, mudstones, limestones and gypsum. Towards the center of the basin, in the most distal areas, mainly lakes, mud flats and salt-pans sedimentary facies associations developed. The evaporitic sequences targeted in this study were deposited in these distal sedimentary environments.

The three large alluvial fans that build up the sedimentary infill of the Loranca Basin have been divided into three stratigraphic units named Lower, Upper, and Final Units (Figure 1). The Lower Unit was deposited during the Upper Eocene-Oligocene, during the initiation of thrusting along the Altomira Range. The Upper Unit includes mostly humid conditions, alluvial fan sedimentary facies at its base (Upper Oligocene-Lower Miocene), which have been described as the First Neogene Unit. Up until this period, the sedimentary sequences that are now isolated within the Loranca Basin were part of a much larger syntectonic Cenozoic basinal area in the center of Iberia, the Madrid Basin ( Vegas et al., 1990; Alonso-Zarza et al., 2004; De Vicente and Muñoz-Martín, 2013). During sedimentation of the top of the Upper Unit, the Loranca basin became endorheic due to its disconnection from the Madrid Basin, related to the emergence of thrusts and formation of a topographic barrier along the Altomira Range (Diaz Molina and Tortosa, 1996). The endorheic sedimentary reorganization was associated with the establishment of much arid conditions in the region, during sedimentation of the Second Neogene Unit sequences. This unit includes four saline/evaporitic sequences





including saline clayey plains and marginal lacustrine environments, well developed in the central part of the Villar de
Cañas Syncline, and that correspond to the area of our tomographic survey. The Final Unit of the Loranca Basin is not
present within the Villar de Cañas Syncline.
In the Villar de Cañas Syncline (Figure 1b), the Cenozoic sedimentary sequences are separated from each other by low
angle unconformities. In particular, the outcropping Lower and Middle Miocene sediments of the Second Neogene Unit
are surveyed by our study (Figure 1c). They are described as the Balanzas series, made up from bottom to top by the
Balanzas Gypsum (Y) and the Balanzas Lower lutites (LT). The Y includes several types of gypsums alternating with
lutites/shales that have been grouped in three units: i) macrocrystalline and laminated gypsums (Y1); ii) gypsum,
shales/lutites and marls (Y2); and iii) gypsum with shaly-marly levels and gypseous alabaster (Ytr) (Figure 1). The LT
crops out in the core of the Villar de Cañas syncline and contains red siltstones and mudstones with some gypsum
and/or sands (Figure 1).
According to the Second Neogene Unit sequence, gypsum rocks are the main lithological target in the study area. The
definition of their internal structure and the boundaries between the different sequences are often difficult, considering
the heterogeneity of these deposits. Gypsum ($CaSO_4 \cdot 2H_2O$) is frequently affected by diagenetic processes and, as a
consequence, gypsum rocks include clay, carbonates and other minerals. The presence of other minerals affects the
purity of the gypsum rocks and this is reflected in changes of its physical properties potentially measurable with
geophysical methods [Carmichael, 1989; Guinea et al., 2010; Festa et al, 2016]. However, their variability in
composition and their complex geometry make the characterization of these deposits challenging [Martinius et al.,
2002; Diaz-Molina and Muñoz-Garcia, 2010; Escavy et al., 2012; Kaufman and Romanov, 2017]., In this case, the
high-resolution seismic characterization of the site was designed taking into account all these structural and lithological
constraints.

**3. DATA ACQUISITION AND PROCESSING**
To provide a detailed image of the target site shallow subsurface, we designed a dense 3D tomographic survey to ensure
a high spatial resolution. The approximately regular acquisition grid covered an area of 500x500m. Source locations
were distributed in a grid of 20x20 m cells. Receivers were distributed along profiles oriented east-west, with 20m
spacing. Along each line, 48 receivers were distributed with a receiver spacing of 10 m (Figure 2). The seismic source
consisted in a 250 kg accelerated weight drop. The seismic data acquisition system consisted in ten 24 channel GEODE
ultra-light seismic recorders (Geometrics systems) that resulted in a 240-channel system. Each channel included a
conventional vertical component geophone. With the available instrumentation, the acquisition scheme required 5
swaths to cover the entire study area.  Each swath consisted in five active receiver lines (240 channels), a total of 676
source shot positions resulting in a total of 3380 shot gathers. The survey was acquired and completed in two different
time periods (December 2013 and January 2014). The acquisition program was carefully adapted to account for the
special circumstances associated to the acquisition of different swath at different times, with different environmental
conditions (e.g. different level of ambient noise, weather changes, or potential technical problems in acquisition
equipment). One of the main concerns was to ensure the releaseof enough acoustic energy for all the available offsets,
especially in presence of complicated weather conditions.  The 250 kg accelerated weight drop source ensured high
signal-to-noise (S/N) ratios in most of the shot points (Figure 3). However, some of them required repettion of the shot
to improve the S/N ratio by means of raw data stacking. Despite the complexity of the seismic acquisition, the recorded
seismic data was of high quality and high S/N ratio and allowed a well-defined picking of first arrivals (Figure 3)
corresponding to almost all the offset range, reaching maximum offsets of almost 700 m. The high quality of the seismic
first arrivals favored the semi-automatic picking of more than a half million of first breaks.
The algorithm used with this data (Pstomo_eq) is a fully 3D traveltime tomographic inversion code [Benz et al., 1996;
Tryggvason et al 2002]. The forward modeling is a first-order finite-difference approximation of the eikonal equation,
computing all the time field from a source (or receiver) to all the cells of the model (two different schemes are available
based on Hole and Zelt, [1995] and Tryggvason and Bergman, [2006]). The traveltimes to all receiver or source
positions are computed from the resulting time field and raytracing is performed backwards, perpendicular to the
isochrons [Vidale, 1998; Hole 1992]. The inversion is performed with the conjugate gradient solver LSQR [Paige and
Saunders, 1982].  One of the main requirements for a successful inversion is the selection of an appropriateinitial model
[Kissling, 1988; 1994]. A good approximation to the minimum starting model is the use of the *a priori* information
available for the area, based on the surface geology and/or the geophysical data previously acquired. In our case, an
initial 1D model was built based on the sonic log information available for different boreholes located within the study
area (Figure 2). The shallow target of the tomographic experiment and the well-controlled sedimentary sequence
expected at these depths, with a non-complex laterally changing geology, favored the election of very suitable initial
models.





The particular acquisition pattern carried out in different swaths forced to establish a careful quality control over the
data. These factors may introduce some bias to the first arrivals picks (Figure 3) that could affect the convergence of the
inversion algorithm. To avoid any potential error associated to the different conditions during the acquisition we decided
to invert all the swaths independently to check the data quality and their convergence. Once convergence was tested in
every swath, the other swaths were gradually added into the inversion. This resulted in a relevant improvement on the
lateral resolution and a better definition of the final velocity model. The inversion of single swaths was also used to test
the dependence of the result on the choice of initial models. Different initial 1D velocity models based on the previous
geophysical and geological information were built to analyse the consistency with the first break picks and the
robustness of the inversion. The best fitting 1D model chosen provided an RMS reduction of 93% showing a clear 2D
trending geometry in the east-west direction. Taking into account this feature, that was also observed in the surface
geology (Figure 1 and, 2), as well as the borehole information, an  initial 2D velocity model was built. This initial
model was then used to speed up the convergence of the calculations and to reduce the number of iterations needed to
reach the optimal final RMS misfit.
The inversion cell size decreased during the integration of the first arrival picks corresponding to the different swaths.
Due to the sparse distribution of receivers in the north-south direction the cell size corresponding to this direction was
the most sensitive to the addition of new traveltime picks to the inversion. Obviously, the reduction of the inversion cell
size was also relevant to increase the resolution  of the final velocity model, resulting in a final inversion grid spacing of
10x20x5 m (for x, y and z).

**4. RESULTS**
The inverted final velocity model shows a detailed image of the shallow subsurface of the study area (Figure 4). This
model provides the best fitting result featuring a final RMS traveltime residual of 2.5 ms, which is indicative of the
good convergence of the inversion process. According to the raypath coverage obtained during the inversion, the
velocity model retrieves the internal structure of the subsurface with a maximum depth of 120 m (Figure 4), especially
in the central and western sector of the survey. This recovering depth decreases drastically towards the east. This was
partly due to the usual ray coverage decrease close to the boundaries of the survey but also to other different causes. The
lateral changes in surface geology in this sector affected seismic source coupling reducing the overall energy injected in
the subsurface and affecting the seismic source repeatability. This issue hindered the identification of the first arrivals in
a wide range of offsets for the shot points located in the eastern part of the study area (Figure 4). Furthermore, the high
velocity gradient observed at very shallow surface also affects the depth of the traced ray paths.
The direct observation of the 3D P-wave velocity model reveals several interesting features about the shallow
subsurface and its internal structure. The tomographic model shows that the shallowest subsurface (first 5-10 m) is
characterized by a very low seismic velocity. This upper layer seems to have a relatively constant depth for the entire
study area. Beneath, there is a velocity gradient smoother towards the northwest, slightly increasing to the south and
significantly to the east (Figure 4). This effect is remarkable in the northeast corner of the study area, where the velocity
rise from 1000-1200 m/s in the shallow surface to up 4000-4500 m/s in the first 20-30 m (Figure 4). This results in a
wedge geometry of the velocity model, indicating a clear northwest dipping trend of the main structural features.
Another significant result is the lateral changes in velocity observed in the deepest part of the model which could
suggest the presence of lateral lithological changes..
Different checkerboard tests were carried out to estimate the potential resolution of the final velocity models  obtained
in the tomographic inversion (Figure 5).  These sensitivity tests provide a qualitative estimation of the spatial and depth
resolution and the uncertainty of the experimental design used. The main idea is to test how well the acquisition
geometry (distribution of sources and receivers) is able to recover a regular distribution of velocity anomalies.
Checkerboard tests only provide indirect evidence of these measures [Lévêque et al., 1993; Rawlinson et al., 2016].
These tests illustrate where, or what parts of the subsurface models are best resolved.  The information that these tests
reveal is similar to the resolution and covariance matrix measures obtained by other conventional schemes. For
example, covariance matrix methods in LSQR [Yao et al., 1999] give incomplete information, especially when sources
and receivers are located at surface. In this case, the raypaths are strongly dependent upon the velocity gradient which
implies a significant non-linearity [Bergman et al., 2004; 2006]. Keeping that in mind, several tests, including using
different size of the anomalies, were applied to our data. Two different sections, one east-west and one depth slice,
representative of the complexity of the study area, have been selected to illustrate the results of the checkerboard
analysis (Fig. 5). First of all, an east-west section located at the center part of the tomographic 3D volume shows that
the velocity anomalies are retrieved for the first 100 m in almost all the study area, slightly reducing its depth of





penetration close to the eastern sector. This fact was expected, due to the lower quality of the first arrivals of the shots located in this area. The traveltime picking carried out in this area were very limited in offset, which clearly impeded to reach deeper exploration depths. On the other hand, a section at constant depth shows a very homogeneous distribution of the anomalies recovered of the checkerboard test. At 45-50 m depth the resolution analysis assures that the ray coverage is homogeneous and well distributed throughout the entire surface. The least covered area corresponds to the north and the southwest part. Although these areas corresponds to the boundaries of the study areas, where lower resolution is expected, technical problems with the geophone cables forced us to disconnect 24 of the 48 channels in the northernmost receiver line and 12 channels in the southwestern sector in four receiver lines in a row. In spite of these acquisition issues, the checkerboard analysis demonstrates the capability of our experimental system/device to image with sufficient detail the shallow subsurface.

### 4.1 Velocity model interpretation

Despite the velocity model provides a detailed image of the shallow subsurface, a direct geological interpretation is difficult, especially in terms of structural features. Interval velocities from sonic logs are critical for a realistic interpretation of the 3D tomographic model so that the internal structure of the shallow subsurface can be geologically inferred.

As mentioned above, the study area was covered by an extensive borehole drilling campaign (including geotechnical and geophysical boreholes with core sampling) and a very detailed surface geology mapping which provided the necessary information to properly decode the geological meaning of the P-wave velocity model. Within the study area only four geophysical equipped wells were available, and used to guide the interpretation of the velocity model (boreholes: SG-29, SG-30, SG-28 and SVC-6, shown in Figure 6). The sonic logs, the tailings and the core samples were critical in linking the different lithologies to the geophysical responses. The lithology and the tomographic image were linked by correlating the velocity profiles obtained from the tomographic model with those determined from the sonic logs. This correlation required a homogenization so that the scales of resolution of both methods would be comparable. The 3D tomographic images are built as velocity grids with cell dimensions of 10x20x5m (x, y, z). This indicates that the sampling interval in the vertical (z) direction is 5 m, while the sample rate in the z axis of the logs is on the cm scale. Thus, the Vp logs needed to be re-sampled to provide average interval velocities in 5 m intervals, representative of the average lithology within this interval. Two resampling approaches were tested. First the log was averaged using a 5 m averaging window, and then re-sampled in 5 m intervals (Figure 4); window lengths from 2.5 to 10 m were also tested, but provided similar results. A median filter approach, that avoids extreme values, was also evaluated, providing a similar response, so the 5m interval average method was finally selected. This homogenization step assured that scale-lengths of the features observed in both data sets were comparable.

The information derived from the well logs provided constraints to interpret up to nine lithological sub-units (Figure 6). However, the relatively reduced overall depth coverage of the tomographic image makes that only four of these units may be identified in the velocity model (Figure 4, and 6). Characteristic lower and upper limits as well as average seismic propagation velocities for P-waves were estimated for each different lithological unit using the sonic log measurements, resulting in the table scale shown in Figure 6.

The gamma ray logs are the most complete logs in the available boreholes, which makes them crucial to define the first lithological boundary at depth (Figure 6). The velocity data is sparser than the gamma ray data, and only the borehole SVC-6, located in the center part of the study area (Fig. 2), provides an almost complete velocity log as a result of the combination of the downhole data and the sonic log. The analysis of the well data differentiates a first upper layer that according to the core samples corresponds to the Balanzas Lower lutites (LT). This sedimentary rock consists mostly of clay minerals with large openings in their crystal structures, in which K, Th and U fit well. This fact makes the gamma ray measurements ideal for its identification because they are very sensitive to the presence of natural radioactivity. A sudden decrease of the gamma ray values clearly defines the transition to a new lithology (Figure 6). The direct observation of the SVC-6 velocity log shows two well distinguished sections characterized by different seismic velocities. The recorded values are relatively low (< 2000 m/s) for the first 10-12 m, with a sudden increase at this depth up to 2200m/s, keeping this velocity relatively constant until the transition zone (~24 m) (Figure 6). The boundary with the deeper formations is observed in Vp with an gradual increase in the velocity values close to 3000 m/s, that takes place in a fewmeters indicating a smooth transition in terms of velocities. From the log analysis it can also be derived that the thickness of the LT layer is almost constant in north-south direction in the central part of the study area (around 20 m depth in SG-28, SG-30 and SVC-6)), increasing its thickness to 32 m in the western sector in which SG-29 is located. The lack of logging information in the Eastern part of the study area forces the interpretation to rely solely on the information from the surface geology (Figure 2) The geological map shows the presence of this lithological interface located in the eastern sector of the study area, with an approximate orientation N-S being sudden moved to the





West in the middle part of the study area. This interface puts in contact the lutites layer with the next lithology identified in the core samples. This fact suggests that the layers dip gently to the West supporting the wedge geometry observed in the tomographic model and following the regional scale geologic interpretations [Biete et al 2012; Piña-Varas et al., 2013].

Just beneath the LT layer, the core samples show the presence of a gypsum-lutite transition layer of nearly constant thickness in most of the study area (at least where the logging information is available) with a local increase in thickness in the Western sector. This unit, called Ytr, belongs to the Second Neogene unit and is characterized by mainly gypsum with centimetric to metric intercalations of shaly/marly levels. These lithological changes are characterized by a high variability in the recorded sonic and gamma ray log values. The presence of these gypsiferous shales are clearly observed in the gamma ray logs, featuring high peaks related to the shaly intercalations. In the sonic logs, the velocity seems to increase in the upper part of the unit with a decrease that coincides with higher presence of the shaly-marly levels (increase in the gamma ray log): Close to the transition to the next lithology, t he sonic log seems to recover the velocity values observed in the upper part.

A great increase in the velocity together with a sudden decrease in the gamma ray values indicates the transition to a thick lithological sequence of gypsum in depth. In terms of borehole logging several subunits can be inferred according to the different signatures observed. However only two of these gypsum units, defined in the geological setting, can be observed/correlated in the tomographic model taking into account the depth achieved with the acquisition geometry used. The upper unit (Y1) is defined by higher values of seismic velocities (~4250m/s) than the deeper unit (Y2) (~3800m/s).

The identification of the main lithological units by means of the logging data and the core samples provides a solid link between the geology and the physical properties that allow us to lay out a structural interpretation the 3D tomographic volume. The standard deviation of every averaged velocity value was used to estimate a rough velocity range corresponding to each lithology but also provided a measure of the quality of the assigned velocity to the different lithologies. This criterion was then used to correlate each P-wave velocity value of the mesh to the defined lithologies (Figure 7). Looking at the velocities table, the LT layer seems to be the most well-established value, according to the standard deviation obtained, which is the lowest (90 m/s). Nevertheless, that this layer was defined only by using the deeper portion of the log data corresponding to this section, which probably corresponds to the higher velocities for this formation neglecting the low values that should be expected at shallow depths due to weathering, unconsolidation etc. which will significantly increase the standard deviation. This is the case of the Ytr layer which features a standard deviation of 400 m/s which clearly reflects the high variability of the seismic velocities observed in the sonic logs.

This velocity analysis allowed us to re-image the tomographic velocity model, this time assigning the different lithologies to the assigned velocity ranges observed in the well log data (Figure 7 and 8). In this way, we could map the four lithological units within the velocity model, clearly defining their respective boundaries and other structural features which characterize the shallow subsurface of the test site.

## 5. DISCUSSION

The direct observation of the different lithological domains identified in the guided interpretation reveals several interesting general features (Figure 8). The defined upper boundaries feature an undulating character that reveals channel-like structures with an east-west orientation. Note that the sedimentary environment during Upper Oligocene-Lower Miocene was meander set up [Diaz-Molina, 1993]. Furthermore, the LT and the Ytr layers appear to increase their thickness towards the west keeping it constant in the north-south direction (Figure 7, 8 and 9) which indicates that the gypsum layers are dipping towards the west. This coincides with the wedge geometry clearly observed in the tomographic velocity models (Figure 4 and 7). The latter was also suggested by the regional scale geology and other geophysical studies [Biete et al., 2012; Piña-Varas et al., 2013].

In order to validate the accuracy of the logging guided interpretation of the tomographic model, several 2D velocity sections in depth were extracted following different existing east-west and north-south geological profiles. Those selected profiles corresponds to geological cross-sections based on data collected at surface and the interpretation of the core samples. Besides, the interpreted boreholes used in this study were also projected to the closest profiles, thus providing additional information to compare and evaluate the final structural interpretation of the 3D velocity grid.

The definition of the different lithological boundaries was one of the main objectives of this study. In this sense, the resulting images show a general good agreement between the geological cross-sections, the interpreted boreholes and




the tomographic models, in terms of boundary definitions, geometry and depth throughout all the 3D volume (Figure 9). The matching between hard data (surface geology plus well-log data) and soft data (seismic tomography) is very consistent taking into account the different criteria used and resolution to define the lithological changes in depth. The correlation between both interpretations is particularly significant in the central part of the study area, where the lithological boundaries defined in the geological cross-sections even show changes in dip and undulating geometries also retrieved by the seismic velocity models. In those areas, the comparison between the interpreted boreholes projected to the velocity profiles is also in good agreement (Figure 9). Nevertheless several discrepancies are observed in specific areas, specially located on western and eastern ends, affecting to different lithological layers, which need to be addressed in detail to finally validate the tomographic models.

From depth to surface the first units identified are Y1 and Y2 (Figure 8). From the previous geological analysis, this gypsum units are characterized by a complex internal structure with no clear defined boundary, continuous lateral changes and the presence of widely disperse massive gypsum bodies. The tomographic model seems to corroborate this by showing a quite complex distribution of these two units in the 3D velocity volume. Unfortunately, the depth resolution of the tomographic model together with the velocity inversion observed between the Y1 and Y2 (Figure 4) makes very difficult to provide a reliable retrieval of the seismic velocities associated to each lithology. This issue is well described in the literature, such as in the one described in Flecha et al. (2004). These aspects together with the smooth character of the seismic tomography leads to consider these gypsum units as a unique lithology, focusing in the upper boundary definition and avoiding the definition of the complex internal structure. Besides, this objective was beyond the scope of the study and from an engineering point of view, both lithological units can be considered as one unit in terms of mechanical response.

The upper limit between Y1+Y2 (Y) and Ytr is relatively well constrained in almost all the area, especially when compared with the log interpretation. Changes in dip and variable geometries in depth observed in the geological cross-sections are also imaged by the guided interpretation of the 3D velocity model (Figure 9). The well contrasted seismic velocities between both lithologies observed in the well logs help in the boundary definition, although the previously mentioned limitations of seismic tomography for these lithologies makes impossible to reach the seismic velocity ranges expected for the gypsum units according to the logging data (Figure 4). The tomography velocity model suggests the velocity inversion in some of the profiles (i.e. profile c9i in Figure 9). The tentative interpretation of the Y1 (Gypsum 1 in Figure 8) clearly does not correspond to this lithology which should be the unit defining the upper boundary of this gypsum sequence; however it is presented in the results because is indicative of the presence of these isolated massive gypsum bodies that are observed and well defined especially in the eastern sector of the study area, some of them very close to the surface.

Quite different is the LT-Ytr boundary definition which seems to be underestimated in depth location as we move to the western sector of the survey area (Figure 9). Several considerations can be taken into account to understand this mismatch observed between different interpretations. First, this lithological boundary is relatively diffuse because of the presence of the gypsiferous lutites as intercalations distributed within the gypsum rock. This fact is the cause of the appearance of peaks of higher velocity in the sonic logs which are responsible for the increase in the average velocity for the Ytr unit.  Unfortunately, due to the acquisition geometry, the resolution that characterizes the tomographic model is not able to differentiate between these intercalations/interfingering (lenticular shape layers of centrimetric to metric scale) within Ytr  which would have helped to define this boundary in greater detail. As mentioned above, we resampled the velocities from sonic logs to correalte their velocities to the tomography results As a result, the averaged velocity associated to Ytr is characterized with a high standard deviation (Figure 6 and 8). This increases the uncertainty of Ytr identification in the whole tomographic 3D volume which induces some mismatch in the unit identification.

On the other hand, the location of the boreholes used for the guided interpretation of the tomographic model also can account for these observed discrepancies. Most of them (SG-28, SVC-6 and SG-30) are located in the central part of the study area and besides they are aligned in the north-south direction. Thus, the weight of these three boreholes in the estimation of the velocity intervals in both lithologies involved is significant and introduces a bias for the rest of the tomography guided interpretation. According to these wells, the lutites have a quite similar thickness placing the lithological boundary at a relatively shallow depth (around 20 m) compared with the same boundary in the western area which is located at a deeper level. This implies that the velocity derived from the boreholes for the LT layer is most probably underestimated in relation to the expected velocities for this lithology at this part of the survey. The effects of the soil compaction, due to the layer thickening in this sector, could increase the velocity of the lutites at depth. This fact but also the standard deviation associated to the Ytr unit seems to be a strong effect in the delimitation of this upper boundary when moving to the west. Both factors lead to a mismatch between the different interpretations. This can be observed in the velocity sections presented in Figure 9. In the east-west velocity sections  there is also a similar effect, although showing a better match between geological and tomographic delineation of this boundary as we move eastward. The north-south sections show definitive evidence of this.  Profile c-9i presents a good agreement between



both interpretations, note that this profile is practically aligned with boreholes SG-28, SVC-6 and SG-30. Conversely,
section c-8i shows a clear discrepancy since it is located further to the west in relation to the c-9i profile.
The tomographic velocity model  suggests the presence of a shallow weathered layer (warm colors in Figure 4). This
layer is clearly observed on the field, the surface mapping and the core samples recovered in most of the geotechnical
boreholes. These observations show that this very shallow layer have two different lithologies that correspond to lutites
(LT) at the northern and western sector of the study area and also transition gypsum (Ytr) in the eastern sector (Figure 2
and 6). This upper weathered layer seems to be characterized by low velocity values though, from a seismic velocity
point of view, both lithologies are barely distinguishable. Furthermore, the guided interpretation of the tomographic
model is also unable to retrieve this layer basically due to the incompleteness of the sonic logs at shallow depths (only
downhole data in available for SVC-6) (Figure 6). This is specially significant for the weathered Ytr unit which has no
recorded data to estimate its seismic velocity at shallow surface. For this reason, in the guided interpretation this
identified weathered layer has not been considered as a differentiated boundary. However, the surface geology offers a
perfect way to define the boundary associated between both lithologies in this upper weathered layer. Methodologically
it indicates that the direct mapping/correlation between velocity and lithology might not be applicable when the
influence of other factors is relevant. Weathering affects the physical properties of the lithology that is outcropping,
decreasing velocities characteristic of the Ytr to values below 2100 m/s, the upper limit criteria used to identify the LT.
The imaging of the LT-Ytr transition cannot be accomplished using only  the tomographic velocity model, according to
the borehole logging data available. More borehole logging data in representative locations of the velocity model are
needed to better constraint the velocity range assigned to each lithology, which in turn would enable to improve the
velocity ranges and reducing the standard deviations for each unit. A complete sequence for the sonic logs, from surface
to the maximum depth, will be very useful  to further constraint the weathered layer and maybe it could offer a clue to
differentiate at surface lutites from gypsum from a seismic velocity point of view. Nevertheless, seismic velocity alone
seems to have some limitations to clearly define both lithologies or at least there is no a clear and unique distinctive
signature for these two lithologies. For this reason, we believed that adding other physical properties (e.g. resistivity or
porosity) could improve the definition of the LT-Ytr transition.
One of the main concerns is the presence of dissolution cavities within the evaporitic sequence, especially taking into
account the possible host of a singular infrastructure. In this sense, traveltime tomography is very limited in recovering
the location, geometry and velocity values expected for a cavity. Besides this is particularly more difficult if only
surface seismic data are used in the inversion (Flecha et al., 2004). In case of the presence of a cavity, the wavefront do
not propagate through it and the first arrivals are only capable to record the perturbation due to the large velocity
contrast at the edge of the velocity anomaly. Fortunately, the density ray diagrams revealed as an appropriate tool to
define the presence of cavities which is characterized by a very low or a lack of ray coverage. Taking into account this,
the analysis of the ray coverage diagrams derived from the traveltime inversion do not show any evidence of this fact
which implies that no cavities are characterized, at least at decametric scale (Figure 4, 5 and 7). Furthermore, the
extensive borehole campaign carried out on site also showed no evidence of the presence of cavities in the shallow
subsurface.
On the other hand, the presence of potentially active faults in the area is also a main issue in hazard analysis and risk
assessment. For this reason, the study of the presence of any non mapped minor fault and the characterization in depth
of mapped ones was also of interest. The study of instrumental and historical seismicity showed that the area was
tectonically stable with a very reduced amount of seismic events in the area and of very low magnitude. Furthermore
the paleoseismic studies by means of trenches revealed that does not exist any evidence of recent seismic activity
related to any fault system. In the same way, the analysis of the tomographic velocity model supports these statements
about evidences of recent faulting responsible of any seismic activity that it could constitute any risk. The lithological
units imaged by the velocity models no not show any evidence of faulting which indicates that this sedimentary package
has not been affected by any recent activity (Figure 7, 8 and 9). This fact supports the evidences showed by other
studies carried out in the area.
**6. CONCLUSIONS**
The detailed 3D structure of an evaporite sequence in the Villar de Cañas syncline (Cuenca) has been revealed by using
high resolution shallow seismic tomographic inversion of first arrival traveltimes. The local tomographic image of the
evaporite sedimentary sequence allows observing undulating structures in the base of the boundary layers. The
tomographic Vp velocity model interpreted with the aid of additional geological and geophysical observations, such as
Vp measurements from sonic logs and core description from boreholes provided a detailed mapping of the different
lithologies that build up the sedimentary evaporite sequence. Additional constraints coming from sonic and gamma ray





logs were proven to be critical in the interpretation of the inverted velocity model, allowing identification of the detailed features and geological structures at depth. Well logs and surface geology data allowed interpreting the different lithologies in the seismic image. The constraints used consisted in average Vp values and Vp ranges for the different lithologies identified from the description of the core samples extracted from the boreholes. This provided the basis for a pseudo-automatic (geophysically-driven) interpretation, where model cells were assigned to a specific lithology according to the Vp value of the corresponding node of the mesh. Despite the relatively complex structure and composition of the target area, the guided interpretation scheme presented in this study results in a very powerful procedure to extract structural information from velocity models. However, the consistency between the model and interpretation reduces its effectiveness when trying to resolve areas characterized by a high uncertainty in the guided interpretation. This is particularly true for the uppermost layers where discrepancies can be accounted for by different factors including: the irregular distribution of the boreholes and logging information; overlapping Vp values for different lithologies/composition; the influence of physical conditions (pressure, temperature, water content). Therefore, in those areas the direct mapping/correlation between velocity and lithology might not be applicable without the help of other constraints, e.g. other geophysical parameters that can provide additional information to distinguish specific lithologies.

**Acknowledgments**

This work has been supported by projects ref: CGL2014-56548-P, CGL2016-81964-REDE supported by the Spanish Ministry of Science and Innovation, 2017 SGR 1022 Generalitat de Catalunya and, by ENRESA. The seismic data recording system that consisted in 10 GEODE (Geometrics) was provided by the GIPP-GFZ Potsdam (Germany). The acoustic energy used as source was generated by a 250 kg accelerated weight drop provided by the Instituto Superior Tecnico, Univ. Lisbon, (Portugal) and a 90 kg accelerated weight drop provided by the Univ. of Oviedo, (Spain). The data is located at the data base server from the Institute of Earth Sciences Jaume Almera, ICTJA, CSIC (http://geodb.ictja.csic.es/#dades1) We are very thankful to Dr. Christan Haberland, Dr. J.M. Gonzalez-Cortina and Prof. J. Pulgar for their interest in the experiment and assistance during the field operations. We would also like to thank the valuable comments of Dr. Juan Alcalde that significantly improved the final manuscript.



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

**Figure Captions**

Figure 1. (a). Simplified geological map of the Iberian Range in eastern Iberian peninsula, with the location of the study
area marked in black box, (modified from Guimerà et al. (2004). (b) Local geological map of the Villar de Cañas
syncline. The target area is marked by a blue rectangle. The 2D seismic reflection profiles acquired in this experiment
are also located in the map. (c) Detailed stratigraphic columns describing the main units observed in both flanks of the
syncline.

Figure 2.Geometry of the acquisition experiment. Red dots are position of the source, white dots are the position of the
receivers. Receivers consisted in single vertical component exploration geophones connected to an array of 10 GEODE
(Geometrics) data acquisition system. Light blue dots indicate drilled boreholes. Weight drop (250 kg) used (from the
Inst. Superior Tecnico Lisbon, Portugal)

Figure 3. Example of shot gather recorded by the array of 10 GEODES, 24 channels each. The red ticks indicate the
traveltime picks of the firsts arrivals used as inputs for the tomographic inversion.

Figure 4.  (a) 3D Seismic compressional wave velocity model (Vp) derived from the over 500.000 traveltime picks of
the first arrivals in the shot gathers. The velocity range goes from nearly 900 m/s (reds) to over 4500 m/s (blues). (b)
Comparison between the smoothly resampled Vp log derived from the sonic at borehole SVC-6 (light blue) and the
vertical Vp profile extracted from the block at the location of the SVC-6 indicated by a black arrow in the block. The re-
sampling of the log was carried out so that it would be comparable to the grid size used for the parametrization of the
velocity model which in this case is of 10x10x5m.

Figure 5.  Checker-board tests, synthetic recovery test taking into account the real acquisition geometry on a model
involving velocity anomalies of dimensions 75x50x25m, approximately. Checker-board analysis is a conventional
scheme to assess the resolution of the tomographic inversion approaches [Leveque et al., 1993; Rawlinson et al.,
20154]. (a) Corresponds to a cross-section of the input synthetic velocity model consisting of box anomalies. (b)
Corresponds to a cross-section across the recovered velocity model. The best resolution is between 25 and 75 m. depth.
(c) Corresponds to a depth slice (map view) across the input model  showing the synthetic velocity anomalies (squares).
(d) Corresponds to a depth slice across the recovered velocity model at a 50 m depth. (e) Is the acquisition geometry, the
red line near the top corresponds to the location of the depth section of (a) . In the cross sections and depth slices red
corresponds to low seismic velocity and blue to high values of the seismic velocities.

Figure 6. Drill-holes in the target area with the borehole geophysic logs used in this study. This reveals the correlation





between the rock samples, its description and the values of the physical properties, Gama ray (GR), sonic logs (Vp). The top part of the figure reveals the logging data with the correlation between the available boreholes. The bottom table defines the summary criterion used for the interpretation of the different lithologies. The Vp value  should be representative of the corresponding  lithologies. This criterion is used later in the text to differentiate between the different lithologies in the velocity cube obtained from the tomographic inversion. The left box illustrated the location of the boreholes within the acquisition geometry of the seismic survey, with the outcroping geology of the target area.

Figure 7. Velocity model grid of the shallow subsurface, it has been color coded according to the lithologies indicated in the included scale/table which is the one derived in Figure 6. LT, Lutites; Ytr the gypsum-lutite transition layer; Y1 and Y2, Gypsum units.

Figure 8, 3D Volume representations of the different lithologic units. LT, Lutites; Ytr the gypsum-lutite transition layer; Y1 and Y2, Gypsum units.

Figure 9. Resulting shallow subsurface structure represented as detailed cross-sections. Cross-sections integrate the velocity model derived from the tomography, the constraints provided by the boreholes and the extrapolation of surface geology data (in discontinuous drafted lines). Four different east-west and north-south cross-sections are showed with their locations within the study area. Note that towards the east (in the geological map), the Ytr outcrops at surface, as well as the Y unit. This revelas that the entire layered sequence dips to the west.



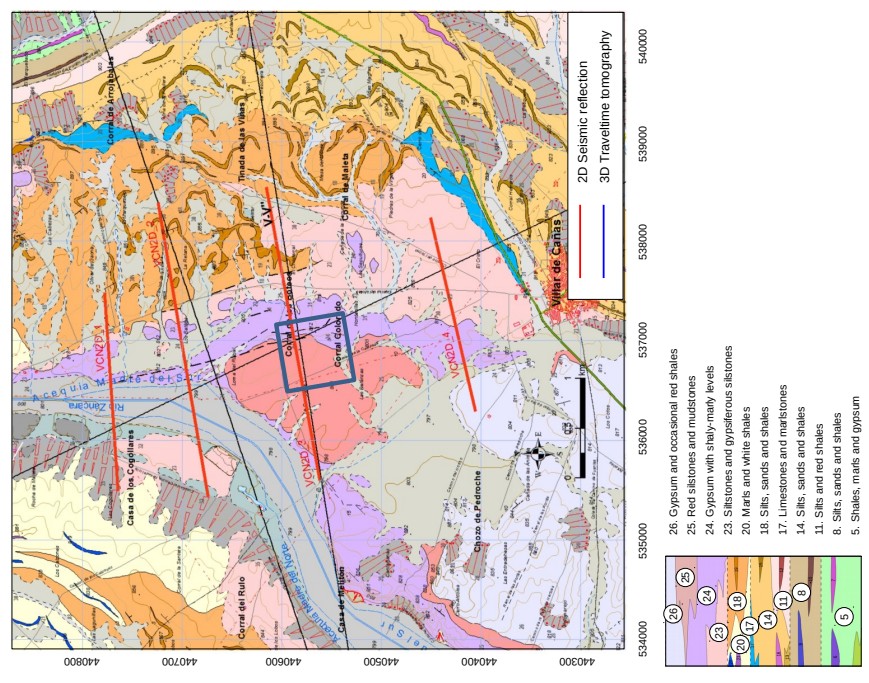

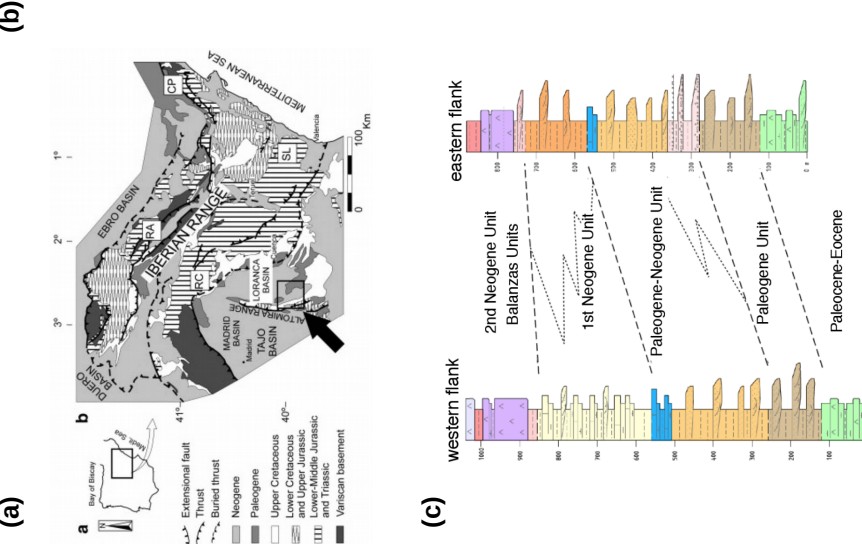

Fig. 1



Fig. 2





Fig. 3





Fig. 4





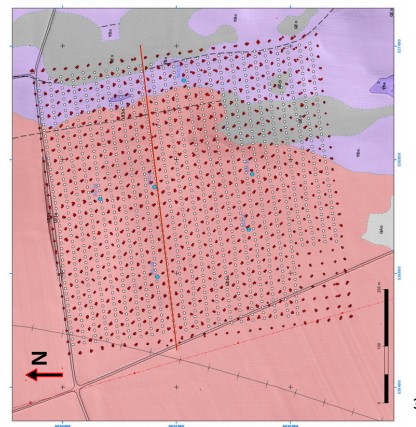

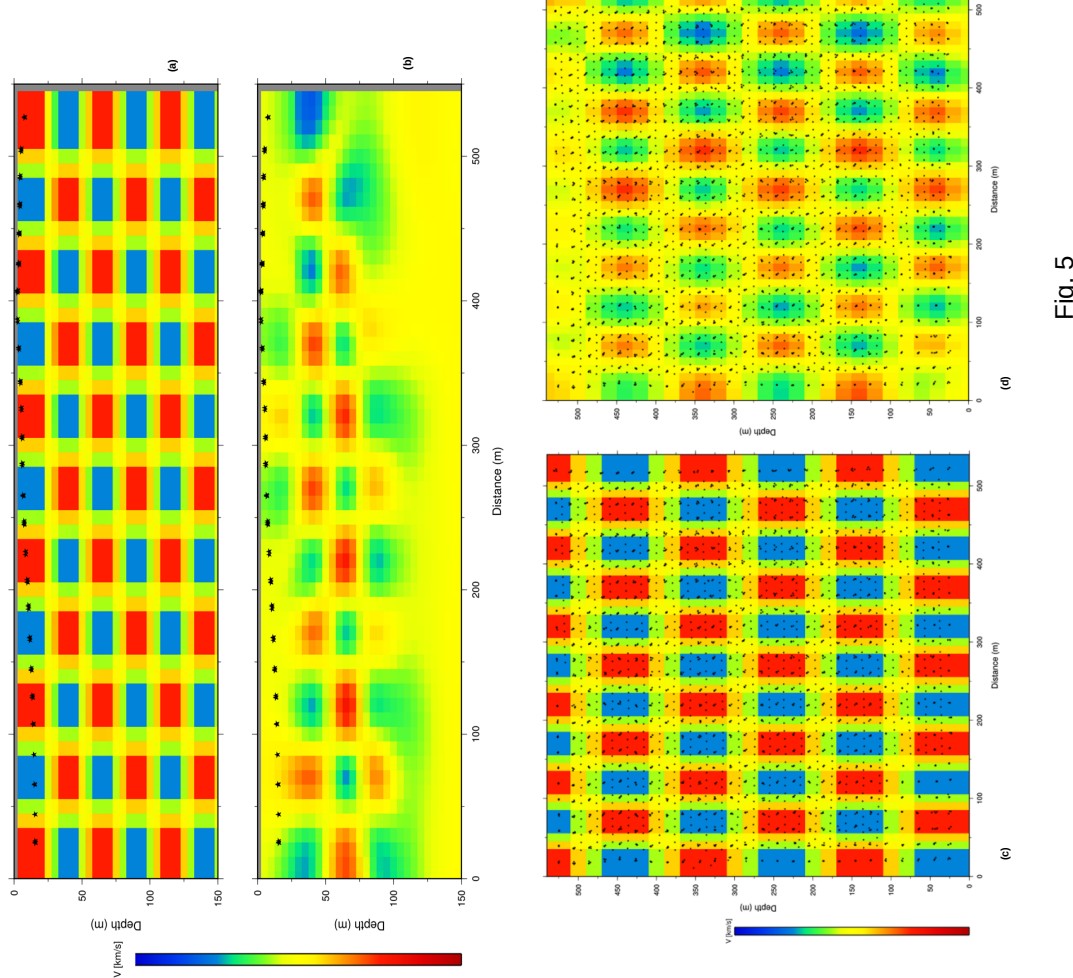

Fig. 5




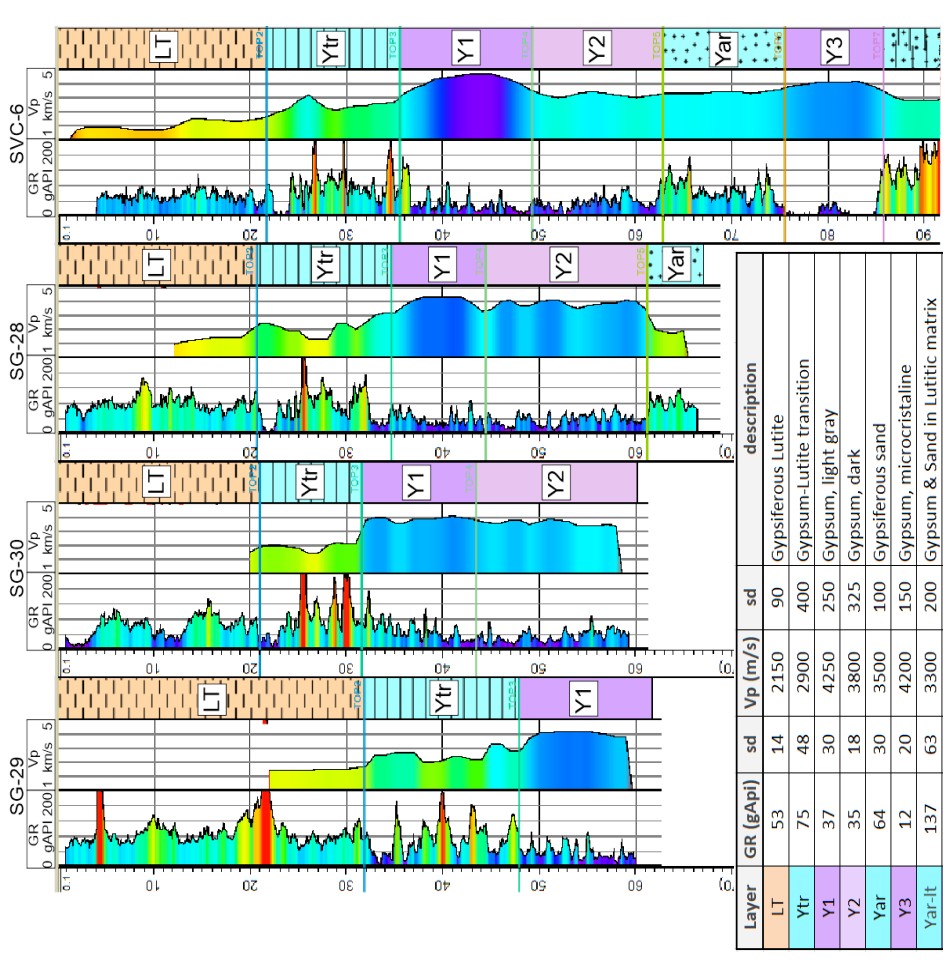

Fig. 6



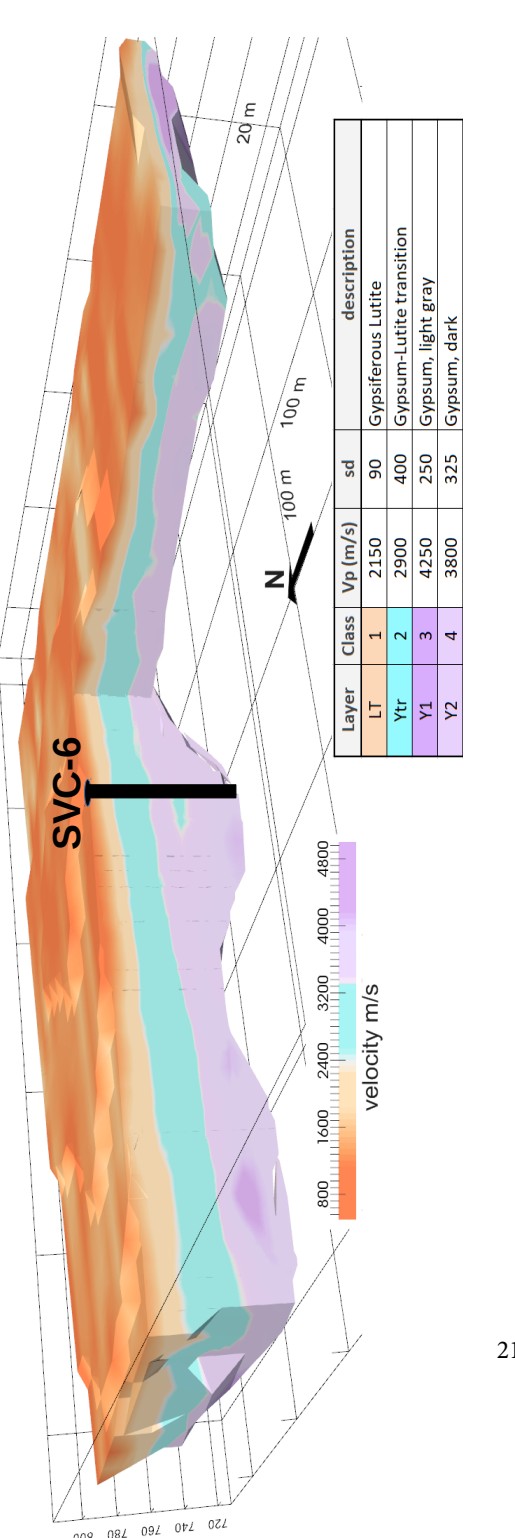

| Layer | Class | Vp (m/s) | sd | description |
|-------|-------|----------|-----|-------------------------|
| LT | 1 | 2150 | 90 | Gypsiferous Lutite |
| Ytr | 2 | 2900 | 400 | Gypsum-Lutite transition |
| Y1 | 3 | 4250 | 250 | Gypsum, light gray |
| Y2 | 4 | 3800 | 325 | Gypsum, dark |

Fig. 7

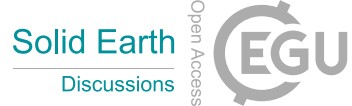



| Layer | Class | Vp (m/s) | sd | description |
|-------|-------|----------|-----|-------------|
| LT | 1 | 2150 | 90 | Gypsiferous Lutite |
| Ytr | 2 | 2900 | 400 | Gypsum-Lutite transition |
| Y1 | 3 | 4250 | 250 | Gypsum, light gray |
| Y2 | 4 | 3800 | 325 | Gypsum, dark |

Fig. 8





Fig. 9