# Peer review of "3D Seismic Traveltime Tomography Validation of a Detailed 1 Subsurface Model: The case study of the Zancara River Basin 2"

_Solid Earth, 2018_

## Referee Comment (RC1) · Dr. Jaiswal (Referee) · 13 Sep 2018

General comments - The paper uses a dense traveltime refraction dataset to map lithology. Model development part of the paper is strong. Quantitative interpretation part of the paper (Fig. 6 onwards) is weak.

Specific comments - The Vp - lithology relation has been built using a series of logs. This is not wrong, just limited in its scope. Logs have a higher resolution than the Vp model. To reconcile logs with the traveltime Vp, authors have averaged the logs within a window and resampled it again. This is a good qualitative approach, not quantitative. Running average is not the same as Backus average, but this is just a minor issue.

[Figure]

Authors will appreciate averaging creates uncertainty. The sense of uncertainty in quantitative interpretation is missing.

I suggest approaching quantitative interpretation in one of two ways. Either, develop a rock physics model for individual lithologies or present Fig. 7 - 9 in a probabilistic sense (what is the probability of a certain point in space to correspond to a certain lithology). Authors have everything they need for both approaches.

- All the best

---

## Referee Comment (RC2) · M. Majdanski (Referee) · 14 Sep 2018

General comments

The manuscript presents an analysis of very dense 3D seismic dataset to characterize the shallow subsurface. The authors suggest that use of a simple first-breaks traveltime tomography method can lead to high-resolution image of the studied area. I have already seen presented data and an initial interpretation at a conference presented by Ramon Carbonell. The data are indeed of high density and good quality, and supported with some boreholes information, so reflection imaging should be a natural interpretation technique. Unfortunately, authors do not discuss this possibility focusing on the

simplest possible technique, claiming that reflection imaging is not needed.

In the first part of the manuscript authors present clear description of traveltime inversion supported with the uncertainty analysis (checkerboards). In the second part they try to combine smooth velocity model from traveltime tomography with sparse borehole data. This is very difficult and questionable part, and would be much easier with reflection imaging.

The manuscript is well written, and easy to read and understand. In the second part it is also a bit long and unconvincing.

Specific comments

Use of the first breaks only. In the near-surface data the first breaks are difficult to pick, especially in the near field. What was the data quality for those short offsets? Fig.3 present only mid-offset arrivals. What data processing (filters, agc) has been applied to the data. What is presented in Fig.3, it is definitely not raw recordings

Line 170 – what do you mean by suitable did you mean "very smooth initial model"? Line 211: it is more like in the first 40 m, as shown in Fig.4, not 20-30

Figure 5 caption – there is no need to explain what is the checker-board test in the figure caption. It is described in the text. Also references to methodology in figure caption is unnecessary. Moreover, reference to Rawlinson et al. 20154 (should be 2014) is missing

Rawlinson, N. and Fichtner, A. and Sambridge, M. and Young, M.K. (2014) Seismic Tomography and the Assessment of Uncertainty. In: Advances in Geophysics. Andvances in Geophysics, 55 . Elsevier, pp. 1-76. ISBN eBook ISBN: 9780128003718 Hardcover ISBN: 9780128002728

Figure 5 – a-e marks and all labels are tiny; values in colour scale are missing; what are those black dots in a and b? What parameters (velocity in anomaly) has been used for checkerboard test?

The checkerboard results shows that the first 25 m has not been resolved at all. I assume this is because of usual problem with picking the first breaks at very short offsets in the near field. In the article no short offset arrivals are presented. Could you comment on that?

4.1 Is this sub section necessary? There is no 4.2

Profile c-2i in figure 9 shows no correlation between tomographic velocity and extrapolated surface geology. The only consistency might be found with ray penetration. I would avoid writing "matching is very consistent" (in line 348)

Figure.1 b small inlet has tinny font, but there is space to change legend to two column text

Figure.2 a b c marks are not needed, as they are not mentioned anywhere

Figure.3 what processing has been applied to this data? No information in caption nor in text

Figure 9 half of markings are tiny

Minor editorial corrections

Line 86 – double dot (.)

Line 133 – unnecessary comma after .

Line 151 – missing space "releaseof"

Line 157 – of first breaks -> of the first breaks

Line 441 – taking into account this -> taking this into account

Line 451 - that does not exist any evidence of recent -> that there is no evidence of recent

Line 454 – models no not show -> models do not show

Line 724 – m. -> m

---

## Author Comment (AC2) · 9 Nov 2018

Dr. Jaiswal (Referee) priyank.jaiswal@okstate.edu

- General comments - The paper uses a dense traveltime refraction dataset to map lithology. Model development part of the paper is strong. Quantitative interpretation part of the paper (Fig. 6 onwards) is weak. Specific comments - The Vp - lithology relation has been built using a series of logs. This is not wrong, just limited in its scope. Logs have a higher resolution than the Vp model. To reconcile logs with the traveltime Vp, authors have averaged the logs within a window and resampled it again. This is

a good qualitative approach, not quantitative. Running average is not the same as Backus average, but this is just a minor issue.

Authors will appreciate averaging creates uncertainty. The sense of uncertainty in quantitative interpretation is missing. I suggest approaching quantitative interpretation in one of two ways. Either, develop a rock physics model for individual lithologies or present Fig. 7 - 9 in a probabilistic sense (what is the probability of a certain point in space to correspond to a certain lithology). Authors have everything they need for both approaches. - All the best

One of the main goals of this manuscript was to test the capabilities and the limitations of the guided interpretation of the tomographic model to define structural features such as lithological boundaries. This guided interpretation was mainly based on logging data which was used to define the different lithotypes to be upscaled to the 3D tomographic velocity model. Unfortunately, one of the main problem was the incompleteness of the velocity logs in most of the boreholes (except SVC-6) in their upper part. This fact basically affected the gypsiferous lutites (LT) underestimating the velocity range given to this lithology and introducing a high ambiguity in the LT-Ytr boundary definition but also to differentiate . In addition to that, the bias imposed by the boreholes, located mainly in the central part of the survey, had also a significant impact in the velocity range definition and mismatch observed between the geological interpolated model and the tomography.

For these reasons, the guided interpretation presented in this manuscript is following a qualitative approach. We believe that a rigorous uncertainty analysis is difficult to carry out taking into account the inconsistency in the velocity log data and this approach is beyond the scope of this manuscript. We suggest to modify the figures showing the guided velocity interpretation in order to emphasize those ambiguity areas (velocity ranges) that have not been assigned to any of these upper lithologies (LT and Ytr).

---

## Author Response (AR1)

**REVIEWER 1:**

**Dr. Jaiswal (Referee)**
**priyank.jaiswal@okstate.edu**

*General comments - The paper uses a dense traveltime refraction dataset to map lithology. Model development part of the paper is strong. Quantitative interpretation part of the paper (Fig. 6 onwards) is weak. Specific comments - The Vp - lithology relation has been built using a series of logs. This is not wrong, just limited in its scope. Logs have a higher resolution than the Vp model. To reconcile logs with the traveltime Vp, authors have averaged the logs within a window and resampled it again. This is a good qualitative approach, not quantitative. Running average is not the same as Backus average, but this is just a minor issue.*

*Authors will appreciate averaging creates uncertainty. The sense of uncertainty in quantitative interpretation is missing. I suggest approaching quantitative interpretation in one of two ways. Either, develop a rock physics model for individual lithologies or present Fig. 7 - 9 in a probabilistic sense (what is the probability of a certain point in space to correspond to a certain lithology). Authors have everything they need for both approaches.*
*- All the best*

The main goal of this manuscript was to reveal the 3D distribution of seismic velocities and to test the capabilities and the limitations of the guided interpretation of the tomographic model to define structural features such as lithological boundaries. This guided interpretation was mainly based on logging data which was used to define the different lithotypes to be upscaled to the 3D tomographic velocity model. Unfortunately, one of the main problem was the incompleteness of the velocity logs in most of the boreholes (except SVC-6) in their upper part. This fact basically affected the gypsiferous lutites (LT) underestimating the velocity range given to this lithology and introducing a high ambiguity in the LT-Ytr boundary definition but also to differentiate. In addition to that, the bias imposed by the boreholes location, most of them located in the central part of the survey, had also a significant impact in the velocity range definition and mismatch observed between the geological interpolated model and the tomography. The fact of the layers are dipping to the west impose changes in velocity in this direction with increasing depth, being the shallow lutites the most affected lithology. The lack of wells in this area cannot account of these changes in physical properties of this lithology.

For these reasons, the guided interpretation presented in this manuscript is following a qualitative approach. We believe that a rigorous uncertainty analysis is difficult to carry out taking into account the inconsistency in the velocity log data and this approach is beyond the scope of this manuscript. To show the ambiguity associated to the boundary definitions we changed the re-imaging of the final velocity model introducing the gaps between velocity ranges which corresponds to seismic velocities that could not be assigned by the well log interpretation. This approach simplifies the understanding of the paper discussion and better supports it.

**Reviewer 2**

**M. Majdanski (Referee)**
**mmajd@igf.edu.pl**

*General comments*

*- The manuscript presents an analysis of very dense 3D seismic dataset to characterize the shallow subsurface. The authors suggest that use of a simple first-breaks traveltime tomography method can lead to high-resolution image of the studied area. I have already seen presented data and an initial interpretation at a conference presented by Ramon Carbonell. The data are indeed of high density and good quality, and supported with some boreholes information, so reflection imaging should be a natural interpretation technique. Unfortunately, authors do not discuss this possibility focusing on the simplest possible technique, claiming that reflection imaging is not needed.*

*In the first part of the manuscript authors present clear description of traveltime inversion supported with the uncertainty analysis (checkerboards). In the second part they try to combine smooth velocity model from traveltime tomography with sparse borehole data. This is very difficult and questionable part, and would be much easier with reflection imaging.*

*The manuscript is well written, and easy to read and understand. In the second part it is also a bit long and unconvincing.*

The results presented are part of a more extensive seismic acquisition experiment carried out in Villar de Cañas (Spain). The experiment included four 2D seismic reflection profiles (Figure 1 in the manuscript) and also the 3D seismic survey presented in this manuscript. The acquired data aimed different targets according to the interests of the contractor company. The 2D seismic profiles were designed to obtain a more structural regional information focused on fault network. On the other hand, the 3D seismic survey was planned to characterize the very shallow subsurface at the future storage facilities location. The main objective of this survey was the distribution of the physical properties of the very shallow surface including the definition of the upper lithological contacts (only the upper sedimentary sequence corresponding to the 2nd Neogene Unit) which directly interact with the ongoing construction works. Taking into account the constraints imposed by the approved project itself we designed the most dense 3D seismic survey possible with different data processing options. In this manuscript we only present the results derived from the travel time tomography which pursued to provide a 3D distribution of physical properties of the very shallow surface and its geological interpretation. This is very valuable for the company to correlate with other physical properties obtained by means of other geophysical techniques (for instance, electrical resistivity tomography) that can be very useful to define the needs in the future construction process. In this manuscript we only focus on velocity models and we also tried to provide structural information using the join interpretation with borehole data and geological mapping.

According to the reviewer comments, it seems that it is not clearly stated in the manuscript. The authors do not want to claim the seismic reflection is not needed and we agree that this method is the most suitable for the structural imaging. The presented results show the capabilities and the limitations of traveltime tomography to image the evaporite sedimentary sequences in a very shallow location. The interest in obtaining a 3D distribution of the very shallow subsurface physical properties of the site and the need of obtain a geological interpretation of this volume was the main motivation of this paper. For this reason, we have changed the abstract and the introduction of the manuscript to clearly state the objectives of the paper and to avoid

***Specific comments***

*- Use of the first breaks only. In the near-surface data the first breaks are difficult to pick, especially in the near field. What was the data quality for those short offsets? Fig.3 present only mid-offset arrivals. What data processing (filters, agc) has been applied to the data. What is presented in Fig.3, it is definitely not raw recordings.*

The data quality was also pretty good for short offsets. Every shot point was shot five times with different geophone lines configuration. The closest offsets obtained were around 10 m (shot points close the active geophone line). For this reason, most of first arrivals corresponding to short offset are pretty clear and it was possible to include them in the tomographic inversion.

For display purposes and only for that, a trace balancing of 1500 ms was applied to the data. No band filters or any other processing was applied to the data showed in Figure 3.

*- Line 170 – what do you mean by suitable did you mean "very smooth initial model"?*

The existing equipped boreholes and the detailed surface geology carried out in the site, previous to our seismic acquisition, made really easy to built a proper initial tomographic velocity model. This was essential to achieve a very fast convergence of the tomographic inversion. For this reason we changed "suitable" for "proper" to provide the correct meaning to the sentence.

*- Line 211: it is more like in the first 40 m, as shown in Fig.4, not 20-30*

This sentence is referred to the eastern sectors of the survey. This can be easily observed in Figure 9 in the EW profiles (C-2i and C5i). The gypsum is outcropping in these areas and the transition between the surface weathered layer to most consolidated gypsum is taken place in 20-30 m with a very high velocity gradient.

*- Figure 5 caption – there is no need to explain what is the checker-board test in the figure caption. It is described in the text.*

We removed the checker-board explanation from the figure caption

*- Also references to methodology in figure caption is unnecessary. Moreover, reference to Rawlinson et al. 20154 (should be 2014) is missing*
*Rawlinson, N. and Fichtner, A. and Sambridge, M. and Young, M.K. (2014) Seismic Tomography and the Assessment of Uncertainty. In: Advances in Geophysics. Andvances in Geophysics, 55. Elsevier, pp. 1-76. ISBN eBook ISBN: 9780128003718 Hardcover ISBN: 9780128002728*

Reference has been removed from the figure caption.

*- Figure 5 – a-e marks and all labels are tiny; values in colour scale are missing; what are those black dots in a and b? What parameters (velocity in anomaly) has been used for checkerboard test?*

The figure has been modified according to the comments of the reviewer. The size of the labels are bigger than before. The values of the color scale have been added to the figure. Those values show percentage of velocity perturbation. The "black dots" corresponds to the location of the shot points . This provides a reference to check the checkerboard recovery in depth. This information has been included in the figure caption.

*- The checkerboard results shows that the first 25 m has not been resolved at all. I assume this is because of usual problem with picking the first breaks at very short offsets in the near field. In the article no short offset arrivals are presented. Could you comment on that?*

The software used do not allow to introduce the topography as a reference for the velocity grid. Therefore the inversion grid is a regular cube referred to a maximum elevation. Those areas above the topography line (black dots in the checkerboard test) are not covered by ray path and thus no anomaly recovery can be obtained. This is specially significant in the western part of the survey in which lower elevation is observed. The velocity model located above the topography have been covered and some explanations have been added to the figure caption.

*- 4.1 Is this sub section necessary? There is no 4.2*
Sub section 4.1 has been removed in the manuscript and it has been included directly to Results section.

*- Profile c-2i in figure 9 shows no correlation between tomographic velocity and extrapolated surface geology. The only consistency might be found with ray penetration. I would avoid writing "matching is very consistent" (in line 348)*
The sentence has been changed to "matching is quite consistent". The lower boundary between transition layer to gypsum is very consistent in all the area, even in profile c-2i. The mismatch occurs between the lutites layer and the transition that is difficult to constraint due to several reasons explained in the manuscript.

*- Figure.1 b small inlet has tinny font, but there is space to change legend to two column text*
The size of the legend has been increased in the geological map

*- Figure.2 a b c marks are not needed, as they are not mentioned anywhere*
The a,b and c marks have been removed and the labels of the location map are now bigger than the previous version.

*- Figure.3 what processing has been applied to this data? No information in caption nor in text*

Trace balancing was applied to the data for display purposes only. This information has been added to Figure 3 caption.

*- Figure 9 half of markings are tiny*
Figure 9 has been modified according to the reviewer suggestions

**Minor editorial corrections**
*- Line 86 – double dot (.)*
Dot removed from the text

*- Line 133 – unnecessary comma after .*
Comma removed from the text

*- Line 151 – missing space "releaseof"*
The space was included in the manuscript

*- Line 157 – of first breaks -> of the first breaks*
Changed in the manuscript

*- Line 441 – taking into account this -> taking this into account*
Changed in the manuscript

*- Line 451 - that does not exist any evidence of recent -> that there is no evidence of recent*
Sentence changed in the manuscript

**LIST OF RELEVANT CHANGES:**

- ABSTRACT and INTRODUCTION: The abstract and the Introduction have been modified slightly to clearly state the objectives of the manuscript, according to the comments of reviewer 2.

- RESULTS: The last paragraphs of the results section have been modified to explain the changes made in the presentation of the guided interpretation of the velocity models.

Related to that, Figure 7 and 9 has been modified according to the new guided interpretation. Figure 8 has been removed of the manuscript because with this new guided interpretation is not offering valuable additional information to the manuscript.

-DISCUSSION: The discussion has been rewritten according to the new figures. The discussion remains unaffected but the new figures better support the arguments used in this section

*Figure changes*

*- Figure.1 b small inlet has tinny font, but there is space to change legend to two column text*
The size of the legend has been increased in the geological map

*- Figure.2 a b c marks are not needed, as they are not mentioned anywhere*
The a,b and c marks have been removed and the labels of the location map are now bigger than the previous version.

*- Figure.3 what processing has been applied to this data? No information in caption nor in text*

Trace balancing was applied to the data for display purposes only. This information has been added to Figure 3 caption.

Figure 7 and 8 have been modified and also the figure captions

Figure 8 has been removed

Figures 9 is Figure 8 in the new version of the manuscript

*- Figure 9 half of markings are tiny*
Figure 9 has been modified according to the reviewer suggestions

*Minor editorial corrections*
*- Line 86 – double dot (.)*
Dot removed from the text

*- Line 133 – unnecessary comma after .*
Comma removed from the text

*- Line 151 – missing space "releaseof"*
The space was included in the manuscript

*- Line 157 – of first breaks -> of the first breaks*
Changed in the manuscript

*- Line 441 – taking into account this -> taking this into account*
Changed in the manuscript

[revised manuscript text omitted]